# Dual-Pathway Neural Networks: Harnessing Scene and Object Pathways for Enhanced Visual Understanding

## Abstract

Standard artificial neural networks (ANNs) often struggle with generalization due to their reliance on surface-level cues, which can lead to suboptimal performance. Drawing inspiration from the distinct processing pathways for scenes and objects in the human brain, we explore the interactions between scene and object and introduce a dual-modality architecture aimed at emulating this cognitive processing mechanism within ANNs. Our approach features separate encodings for scene and object modalities, which are fused to facilitate enhanced visual understanding. By optimizing object recognition and scene reconstruction objectives, our architecture efficiently encodes scene and object information crucial for holistic representation learning. Empirical validation demonstrates significant improvements in generalization, lifelong learning, and adversarial robustness compared to conventional architectures. These findings underscore the potential of integrating biological insights into AI systems to bridge the gap between artificial and biological intelligence. [1]

## 1 Introduction

Artificial neural networks (ANNs) have made significant strides in mimicking human-like intelligence, achieving remarkable performance across various vision tasks (Guo et al., 2016; Arani et al., 2022; Islam et al., 2023). However, despite their successes, contemporary ANNs exhibit notable limitations that impede their ability to emulate the robustness and adaptability inherent in human visual perception. Standard DNNs are vulnerable to shortcut learning (Shah et al., 2020) and adversarial attacks (Carlini et al., 2019), and they are more biased towards texture (Geirhos et al., 2018), latching on to superficial cues rather than a robust understanding of the objects. This leads to poor generation when there is a domain shift from source distribution (Zhou & Feng, 2018). Furthermore, they fail to adapt to changing environments and suffer from catastrophic forgetting when trained on a continuous stream of data (Parisi et al., 2019). These limitations underscore the need for a deeper understanding of the underlying principles governing human intelligence, as embodied by the intricate workings of the human brain. In particular, ANNs frequently exhibit a propensity to latch onto superficial features, overlooking the deeper semantic context of the visual scene. This reliance on surface-level cues renders ANNs less adept at discerning subtle variations in shape, texture, and context, thereby limiting their ability to learn robust and reliable representations of objects in the real-world.

Notably, ANNs tend to rely more on scene and texture information (Geirhos et al., 2018) than intrinsic characteristics and structure of individual objects. This disparity in object-centric versus scene-centric processing contrasts sharply with the human brain's innate predisposition towards object-based recognition, wherein objects are perceived and understood based on their intrinsic properties and spatial relationships within the visual field (Ishai et al., 1999; Contini et al., 2020). The discrepancy between ANNs and the human brain in this regard underscores the importance of understanding and identifying the cognitive mechanisms underpinning human visual perception, with the aim of informing the design of more biologically inspired and cognitively plausible systems.

---

[1]The code and dataset will be made publicly available upon acceptance.

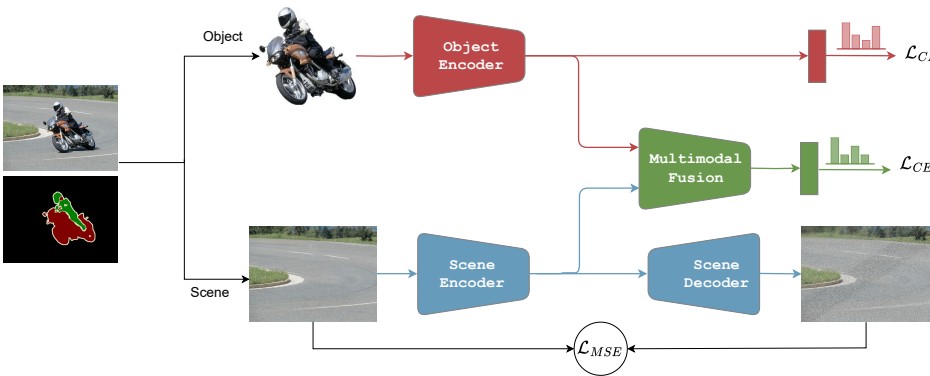

Figure 1: DualPath employs separate pathways to encode object and scene information, which are then fused together in a complementary manner. Additionally, we add auxiliary losses on object and scene encoders to encourage the model to learn semantically meaningful representations. The fusion module and joint optimization of these objectives enable complementary learning and distillation of knowledge between modalities.

Central to the remarkable efficiency of human visual perception is the brain's intricate organization, which incorporates distinct neural pathways dedicated to processing scenes and objects (Nassi & Callaway, 2009; Peelen et al., 2024). Concretely, Peelen et al. (2024) posit that visual object and scene processing occur in parallel, enabling a rapid initial understanding of both concurrently. Furthermore, there is evidence for bidirectional interactions between object and scene processing, where scene information influences object perception and vice versa. Such dual-pathway architecture enables the brain to simultaneously extract both global contextual information and fine-grained object details, facilitating the acquisition of robust and holistic representations of the visual world. By segregating scene and object processing into specialized neural circuits, the brain can disambiguate objects within their surrounding context and infer contextual cues from the scene, enabling a holistic understanding of the environment. Importantly, this integrated processing framework enables the human brain to learn robust object representations and adapt to novel situations. It also allows the brain to disambiguate objects using cues from the scene and vice versa (Peelen et al., 2024). Inspired by this fundamental organization of the human brain, we hypothesize that designing ANNs with analogous distinct pathways for scene and object that inform each other can be beneficial for enhancing the generalization capabilities and robustness of ANNs in complex visual tasks.

To this end, we introduce a dual-modality architecture, DualPath, which aims to disentangle object from scene and processes them using distinct pathways which share complementary information (Figure 1). DualPath leverages separate encodings for scene and object information, which are subsequently fused in a complementary manner to facilitate robust and holistic visual understanding. Specifically, our architecture takes as input paired object and scene images, allowing for the simultaneous extraction of both local object features and global contextual information. To ensure the semantic richness of the learned encoding, we incorporate specialized modules at each processing stage. For object encoding, we employ a classifier tasked with object recognition, guiding the network to capture semantically rich and discriminative object features essential for accurate classification. Conversely, for scene encoding, we utilize a decoder network trained to reconstruct the input scene image, thereby encouraging the network to capture meaningful scene semantics and contextual relationships. The multimodal fusion takes as input the object and scene encodings and fuses them together in a complementary fashion to combine the two modalities. These multimodal representations are guided by the fusion classifier, which is trained with object classification loss. By jointly optimizing these complementary objectives, our architecture enables the efficient encoding of scene and object information, and distills information between modalities, facilitating the extraction of robust and contextually informed representations.

To empirically validate our hypothesis and explore the potential of having distinct pathways for scene and image similar to the brain and allowing interactions between them, we conduct extensive experiments using datasets containing object segmentation masks to extract objects from images and

perform inpainting on the background to create scene images. Our experimental results demonstrate a significant improvement in generalization performance compared to conventional single-pathway architectures. By explicitly modeling separate pathways for scene and object processing, our approach effectively mitigates scene bias and enhances the model's ability to focus on the object and generate object-centric predictions, aligning more closely with the characteristics observed in the human brain. Moreover, we find that the dual path approach enhances the model's lifelong learning capabilities and boosts adversarial robustness, further underscoring its potential to address several shortcomings of standard ANNs. Our study provides early evidence and presents a compelling case for designing biologically plausible architectures that can disentangle objects from scenes and process them using distinct pathways to bridge the gap between artificial and biological systems.

## 2 METHODOLOGY

We first provide an overview of the cognitive mechanisms in the human brain that motivates our study and how we aim to emulate the distinctive processing principles observed in human visual perception. Finally, we delve into the formulation details, elucidating the components and mechanisms underlying our proposed approach.

### 2.1 DISTINCT PATHWAYS FOR SCENES AND OBJECTS IN THE BRAIN

The human brain stands as a remarkable model of cognitive prowess, particularly in its ability to construct robust representations of the surrounding environment and comprehend the intricate relationships and interactions between various objects and their correlation with scenes. This inherent capability enables the brain to navigate diverse settings with ease, leveraging contextual cues to anticipate the presence of specific objects and vice versa. Fundamental to this process is the brain's adeptness at distinguishing between objects and scenes, each represented and processed in a distinct manner to facilitate the formation of associations and inferential reasoning.

A recent neuroscience study by Peelen et al. (2024) argues for the existence of separate neural pathways dedicated to processing scenes and objects within the human brain. This organizational framework allows for mutual information exchange between scene and object representations, enabling the brain to make inferences even when one is partially obscured or blurry, leveraging contextual cues from the other. Such interplay between scene and object processing pathways likely forms a cornerstone of the brain's learning machinery, facilitating robust generalization to novel scenarios by enabling the synthesis of contextual and object-specific information in a complementary manner.

By drawing inspiration from these cognitive principles, we aim to distill similar processing mechanisms in ANNs for scene-object disambiguation and contextual inference, thereby enhancing their generalization capabilities and robustness to distribution shifts in the real-world.

### 2.2 DISTINCT PATHWAYS IN ANNS

Standard ANNs lack explicit mechanisms for separate and concurrent processing of scenes and objects, often failing to distinguish between these visual precepts for object recognition tasks. This conflation makes them susceptible to surface correlations and shortcut learning, leading to texture bias and over reliance on scene context that may be unrelated to the target object.

To this end, our study aims to equip ANNs with dedicated processing units for scenes and objects to explore the benefits of having distinct pathways that interact with each other. The proposed framework disentangles the input images into their constituent scene and object components using the paired object masks and inpainting(Suvorov et al., 2022). Note, that these masks do not need to be precise and can also be obtained using foundation models like SAM Kirillov et al. (2023). The extracted scene and object images are processed by separate encoders to extract optimal modality specific features. The individual representations are then fused together to extract multimodal features, allowing the model to encode scene and object information distinctly and learn their correlations in a synergistic manner. To encourage the model to learn semantically meaningful representations of scenes and objects, we introduce additional components to our architecture. For object representation learning, we incorporate an object classifier that makes inferences solely based on object

encoding. Simultaneously, for scene representation learning, we employ an image reconstruction loss, encouraging the model to capture meaningful scene semantics and contextual relationships.

By leveraging these mechanisms and jointly optimizing the components, our approach enables the model to learn semantically rich representations for scenes and objects, which are then fused in a complementary manner to enhance the generalization capability of the model. Our study aims to provide early evidence for the potential of distinct scene object pathways to address some of the fundamental limitations of current ANNs and pave the way for improved performance in various visual tasks that require a nuanced understanding of scene-object relationships.

## 2.3 FORMULATION

Given an image $x$ with corresponding label $y$ and object mask $\Updownarrow_o$, we extract the object image $x_o$ using the object segmentation mask and the scene image, $x_s$, by removing the object and applying inpainting (Suvorov et al., 2022). Our method utilizes separate encoders for scene and object representations, parameterized by $\theta_o$ and $\theta_s$ respectively, to extract the object representations $z_o$ = $E(x_o; \theta_o)$ and scene representations $z_s = E(x_s; \theta_s)$. The object and scene encodings are then fused together using a fusion module parameterized by $\theta_f$ to extract the fused representations $z_e = E(z_o, z_s; \theta_f)$. The fusion mechanism allows the model to capture complementary information from both scene and object representations, facilitating a holistic understanding of the visual context. DualPath involves jointly training the fusion classification loss and auxiliary losses on scene and object, which allows for knowledge sharing between the modalities and enables rich semantically meaningful representations.

**Object Classification:** To facilitate object representation learning, we train an object classifier $F_o$ parameterized by $\phi_o$, which takes as input the object representations, $z_o$. The classifier is trained using a cross-entropy loss, $\ell_{ce}$. The object classification loss is given by:

$$\mathcal{L}_{\text{object}} = \ell_{ce}(F_o(z_o; \phi_o), y) \tag{1}$$

By explicitly optimizing for classification loss solely on object encoding, the object encoder is encouraged to extract discriminative object features essential for accurate recognition without relying on contextual information from scene which can make the model susceptible to surface irregularities and shortcut learning. This also biases the model towards more robust object-centric recognition, similar to the human brain, and extract intrinsic characteristics and structure of the object.

**Scene Reconstruction:** Scenes provide valuation contextual cues to the model for which objects are likely to be present within a given scene and to disambiguate objects when they are obscured or blurry. This requires the scene representations to be semantically meaningful and facilitate capturing the intricate relationships between scene and objects. To this end, we also add an auxiliary loss on scene encodings. The scene representations, $z_s$, passes through scene decoder,$D_s$, with deconvolution layers parameterized by $\phi_s$ to reconstruct the scene image, which is trained using a mean squared loss, $\ell_{MSE}$ defined as:

$$\mathcal{L}_{\text{scene}} = \frac{1}{N} \sum_{i=1}^{N} \|x_s - D_s(z_s; \phi_s)\|^2 \tag{2}$$

By reconstructing the scene images for scene encoding, the model is encouraged to capture meaningful scene semantics and contextual relationships, facilitating a richer understanding of the visual context. Note that we opt to not train an object classifier on top of scene encoding, as often a scene is not unique to a specific object class but rather a group of them. For instance birds and airplane can share similar scene. Hence cross-entropy can lead to noisy associations and prevent the scene encoder from learning generalizable features. The reconstruction loss, on the other hand, enables the model to learn rich generalizable features that can be utilized by the fusion module to learn object scene correlations.

**Fused Representation and Classification:** Central to our approach is the interplay between scene and object encodings to provide a robust and holistic understanding of the visual task and enables disambiguating objects using contextual cues from scene. The scene and object representations are first flattened and fused to form fused representations, $z_f$, which are then passed through a fused

classifier $F_{\text{fused}}$ parameterized by $\phi_f$. For fusion we use a learnable weighted averaging of object and scene representations using attention.

$$z_s = \mathcal{A}_s \cdot z_s + \mathcal{A}_o \cdot z_o \tag{3}$$

where $\mathcal{A}_s$ and $\mathcal{A}_o$ are the learnable attention weights for scene and object encoding respectively and have the same dimension as $z_s$ and $z_o$. This provide a simple and effective approach for combining information from scene and object based on the quality and utility of the signal. Finally, the fusion classifier,$F_{\text{fused}}$, is trained using cross-entropy loss, defined as:

$$\mathcal{L}_{\text{fused}} = \ell_{ce}(F_{\text{fused}}(z_f; \phi_f), y) \tag{4}$$

By jointly optimizing the fused representations and classification, the model learns to integrate scene and object information effectively and utilize the complementary information in the two modalities to improve the generalization of the model. As the fusion module combines the scene and object representations, the fusion loss also creates synergy between the two modalities and guides learning in the scene and object encoder so that information is extracted in a complementary fashion such that the interplay and relation between scene and object representations enable the disambiguation of objects.

**Overall Loss:**    The overall loss is computed as a weighted sum of the object classification loss, the fused classification loss, and the scene reconstruction loss:

$$\mathcal{L} = \mathcal{L}_{\text{object}} + \lambda_{\text{f}} \cdot \mathcal{L}_{\text{fused}} + \lambda_{\text{s}} \cdot \mathcal{L}_{\text{scene}} \tag{5}$$

where $\lambda_{\text{f}}$ and $\lambda_{\text{s}}$ are regularization parameters. By jointly optimizing these components, our model learns semantically meaningful representations for scenes and objects, facilitating a richer understanding of the visual context and improving performance across various computer vision tasks.

## 3  EXPERIMENTAL SETUP

### 3.1  DATASETS

To test our hypothesis, extraction of scene and object components from an image that can be processed with distinct processing pathways. To this end, we use a subset MS-COCO (Lin et al., 2014) and ADE20K (Zhou et al., 2019) datasets to create an object recognition task. From the set of images with corresponding segmentation masks, we create a subset of images that contains only instance(s) of a single object among the selected objects, and the rest of the image is considered a scene. To have a more uniform distribution and remove the effect of extraneous factors, we cap the number of training samples for each object to 500 and use 50 test samples for each object. For Tiny-MSCOCO, we selected 10 classes, which constitute a total of 4286 images and 500 test samples. Not that for Tiny-MSCOCO, the selected samples do not contain any other objects in the scene. ADE20K presents a more challenging dataset as it is primarily for scene understanding and every pixel is associated with an object. We selected 12 object classes that had sufficient samples and similar to Tiny-MSCOCO, we capped the upper sample count to 500 and used 50 test samples for each object class. For examples of the dataset, selected classes, and sample counts See Appendix, Section A. To create the object image, we extract the image pixels with a segmentation mask for the selected class. Please note that there can be multiple instances of an object in an image. For the scene image, we remove the object pixels and then run LAMA inpainting (Suvorov et al., 2022) to create a smooth scene image. Please note that while our study aims to build the case for mimicking the separate pathways for scene and object, and relies on segmentation mask availability, which limits its potential applications, they do not need to be precise and we believe that the necessity for having object masks can be relaxed by using a foundation segmentation model (Kirillov et al., 2023).

### 3.2  EXPERIMENTAL SETTING

For all our experiments, we employ the ResNet18 (He et al., 2016) architecture as the encoder. The initial convolution layer of the encoders uses a kernel size of 7 and a stride of 2, followed by max pooling with a kernel size of 3 and a stride of 2. In our approach, we utilize separate ResNet18 encoders to capture the scene and object modality. Additionally, we reconstruct the scene

image from its representations using deconvolution layers, following a structure similar to that of the encoder. To ensure a fair comparison, we halved the number of channels in the encoders for DualPath, resulting in a comparable total number of learnable parameters. We train the models using the Adam optimizer and employ a cosine annealing learning rate schedule, starting from a learning rate of 1e-3 and decaying to 1e-5 over 100 epochs. To avoid overfitting, we apply the following augmentations: random resize, random horizontal flip, and randomly applied color jitter or grayscale, followed by random rotation up to 20 degrees.

## 4 EMPIRICAL EVALUATION

To assess the benefits of distinct scene and object pathways, we first compare the generalization performance of the model to the standard ANN trained under uniform experimental conditions. For the baseline model, we integrate the object into the inpainted scene image to create a single combined image. In contrast, for DualPath we provide both the object and scene images to their respective encoders.

Table 1: Generalization performance comparison with standard ANN (Baseline). DualPath provides considerable performance gains.

| Method | Tiny-COCO | Tiny-ADE20K |
|---|---|---|
| Baseline | $78.40_{\pm 0.87}$ | $33.00_{\pm 0.50}$ |
| DualPath | $\mathbf{89.13}_{\pm 0.81}$ | $\mathbf{70.67}_{\pm 1.36}$ |

Table 1 shows the remarkable generalization gains achieved by DualPath across the datasets. Notably, we observe over a 200% improvement in performance on the Tiny-ADE20K dataset compared to the baseline model. Note that TinyADE20K presents a particularly challenging object recognition task as in some cases, the object can be very small in the image, making it difficult for standard ANN without being able to distinguish between object and scene. DualPath, equipped with separate pathways for processing objects and scenes, is able to focus on the object instead of latching onto superficial features in the background and use contextual cues from the scene to identify the object in this challenging setting. Tiny-COCO presents a relatively simpler recognition task as each image contains only one object which often occupies a larger portion of the image. Under this setting too, we observe generalization gains.

We believe that the consistent performance gains can be attributed to the following factors: DualPath enables the model to effectively focus on objects, even when they occupy a small portion of the overall image. Additionally, it can effectively leverage scene information to disambiguate objects, particularly in occluded scenarios. These findings underscore the potential benefits of extracting scene and object components and incorporating separate processing pathways for each, allowing for the sharing of complementary information akin to the human brain's cognitive framework.

### 4.1 CONTINUAL LEARNING CAPABILITY

To further investigate the advantages of employing distinct pathways for processing scene and object, we also consider the continual learning (CL) (Parisi et al., 2019) setting where the model is required to learn a sequence of tasks. To this end, we introduce a Class-Incremental Learning (Class-IL) (Van de Ven & Tolias, 2019) variant of the Tiny-COCO and Tiny-ADE20K datasets, where each task introduces two distinct classes, and the model must learn the new classes while retaining previously acquired knowledge. The order of classes in each task follows the order in Figure 6 and 7. Hence we have 5 disjoint tasks for Seq-Tiny-COCO with two object classes each and 6 disjoint tasks for Seq-Tiny-ADE20K. We also provide results for Task-IL whereby the model has access to the task labels at test time. We train the models under Class-IL setting and only at inference use the task label to limit classification within the task logits.

Among the different approaches for CL, Experience Replay (ER) (Riemer et al., 2018) has been shown to be one of the most effective approaches in mitigating catastrophic forgetting under challenging CL scenarios (Farquhar & Gal, 2018). ER involves maintaining a fixed size buffer to store samples of previously learned tasks and interleaving the training of the new task with earlier task samples to approximate the joint distribution. We hypothesize that having separate pathways for object and scene allows the model to learn more robust and generalization features which are less susceptible to forgetting, and also reduces the impact of the domain shift that occurs due to scene changes.

Table 2: Effect of separate pathways on sequential learning of tasks in continual learning under the experience replay framework. DualPath significantly increases the lifelong learning capability of the model.

| Buffer Size | Method | Seq-Tiny-COCO | | Seq-Tiny-ADE20K | |
|---|---|---|---|---|---|
| | | Class-IL | Task-IL | Class-IL | Task-IL |
| 200 | Baseline-ER | $45.60_{\pm2.51}$ | $78.93_{\pm0.64}$ | $20.33_{\pm0.67}$ | $69.17_{\pm0.88}$ |
| | DualPath-ER | $\mathbf{52.87}_{\pm1.63}$ | $\mathbf{81.67}_{\pm4.82}$ | $\mathbf{36.39}_{\pm3.84}$ | $\mathbf{79.11}_{\pm1.83}$ |
| 500 | Baseline-ER | $49.80_{\pm2.09}$ | $80.90_{\pm0.99}$ | $22.11_{\pm1.57}$ | $69.88_{\pm1.55}$ |
| | DualPath-ER | $\mathbf{62.73}_{\pm1.53}$ | $\mathbf{81.73}_{\pm2.83}$ | $\mathbf{42.00}_{\pm1.41}$ | $\mathbf{81.17}_{\pm0.47}$ |

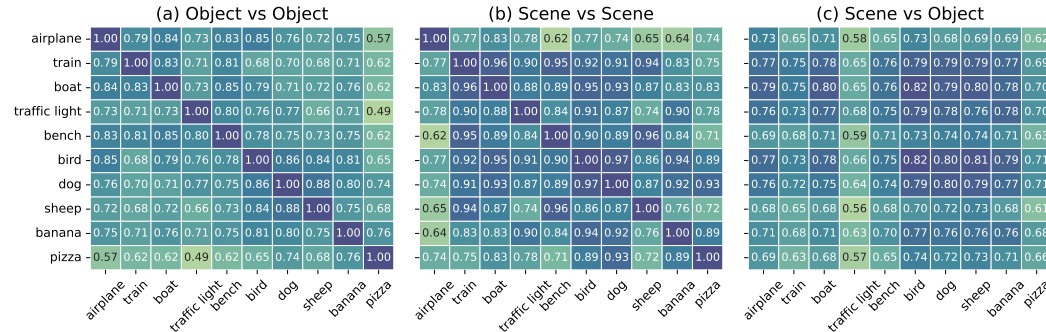

Figure 2: Cosine similarity matrices of the average representations (a) between different objects, (b) between different scenes associated with objects and (c) different objects and scenes.

Table 2 shows that DualPath can effectively mitigate forgetting and significantly enhances the CL capability of the model under different buffer size regimes. We observe manifold better generalization performance and decreased forgetting across tasks. CL remains as one of the key fundamental challenge for ANNs and the considerable gains with DualPath under ER setting compared to standard ANNs provides a compelling case for exploring it further. We believe that having distinct pathways facilitates the learning of more robust and generalizable features which are more robust to distribution shifts. This provides a promising path for AI systems that can seamlessly adapt to evolving environments and tasks.

## 5 EMPIRICAL ANALYSIS

Through a series of experiments and evaluations, we aim to provide insight into the strengths and limitations of our proposed methodology, shedding light on its potential implications for advancing the field of computer vision.

### 5.1 REPRESENTATION SIMILARITY ANALYSIS

To gauge the effectiveness our approach in learning semantically meaningful representations of scenes and objects, we evaluate the similarity between the average class-wise representations of scenes and objects. Figure 2 provides the similarity matrices for objects, scenes and also scene vs object of the same class.

Our analysis reveals that semantically similar objects exhibit high similarity in object representations, indicating the model's capability to capture discriminative features characteristic of each object class. Furthermore, we observe notable similarities between the scenes of objects that commonly co-occur in similar settings, such as benches, birds, and sheep. This observation suggests that the model has successfully learned semantically meaningful representations for scenes too, enabling it to capture and leverage the interactions between scenes and objects in the fused representations.

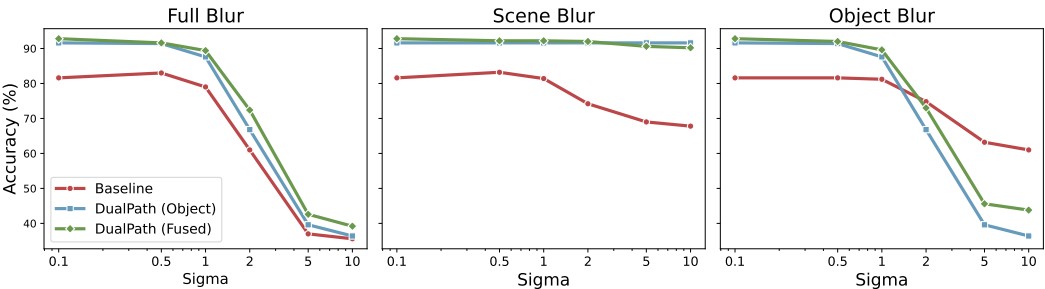

Figure 3: Effect on the performance of the models when Gaussian blur blur is applied to the full image (scene and object), or only pixels corresponding to scene or object.

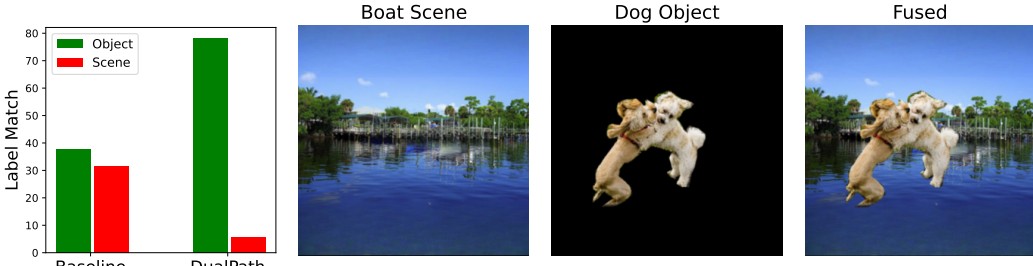

Figure 4: Comparison of the scene bias of the models measured by evaluating the percentage of predictions that match the scene label vs object label.

Additionally, we observe a correlation between objects and scenes of the same image, further underscoring the model's ability to encode contextual relationships between objects and their surrounding scenes. These findings provide evidence for DualPath's capacity to learn rich and contextually informed representations, essential for robust and versatile performance across diverse visual tasks.

## 5.2 ROBUSTNESS TO BLURRING

To evaluate the effectiveness of our approach in utilizing cues from objects and/or scenes for disambiguating blurred images, we systematically apply Gaussian blur with increasing sigma values (using a constant 7x7 kernel) to either the object, the scene, or both. We then compare the impact of this blurring on the model's performance. Figure 3 shows that our approach exhibits considerable robustness to blur in scenes and the entire image. Specifically, we observe that DualPath's performance experiences a relatively lower decrease when the entire image is blurred compared baseline. This suggests that the model is able to leverage contextual information from the scene to compensate for loss of object details, thereby maintaining relatively stable performance. Further, blurring the scene has minimal effect on the performance of DualPath, indicating that the model does not rely on surface level irregularities for object recognition and is hence less vulnerable to shortcut learning.

In contrast, when only the object is blurred, DualPath demonstrates a higher decrease in performance. This result is understandable, as blurring the object directly affects the discriminative features used for object recognition, leading to a more pronounced impact on the model's ability to accurately classify objects. Overall, the analysis shows that DualPath relies more on the characteristics of the objects itself and does not latch onto spurious correlations in the scene.

## 5.3 SCENE BIAS

To assess whether the model relies more on the scene or the object for object recognition, we perform an experiment in which we swap the scene of an image from another object class and evaluate the label match of the model with both the object class and the scene class. Figure 4 shows that DualPath exhibits considerably less reliance on the scene compared to the baseline model. Specifi-

cally, we observe that our model demonstrates higher agreement with the object class compared to the scene class, suggesting that it focuses more on the object to make predictions. In contrast, the baseline model shows high agreement with the scene class, indicating a stronger reliance on scene information for classification. This scene bias in the baseline model highlights its susceptibility to surface correlations and shortcuts, potentially leading to less robust and accurate predictions. Overall, our approach reduces scene bias and enhances the model's ability to focus on the object, leading to improved generalization and reduced susceptibility to surface correlations and shortcuts in the scene.

## 5.4 ADVERSARIAL ROBUSTNESS

As DualPath reduces the susceptibility to superficial features in scenes, we hypothesize that it should also improve the adversarial robustness of the model since the majority of the pixel changes aren't in the object region. To test this hypothesis, we compare the robustness of the models to the more plausible blackbox attack where the adversary creates adversarial examples (Goodfellow et al., 2014) for a surrogate model and does not have access to the gradients of target model. The adversarial examples are created for a standard ANN using a 10 step projected gradient descent attack (Madry et al., 2018) with 0.03 step size and epsilon of 6/255 and tested on baseline model and DualPath.

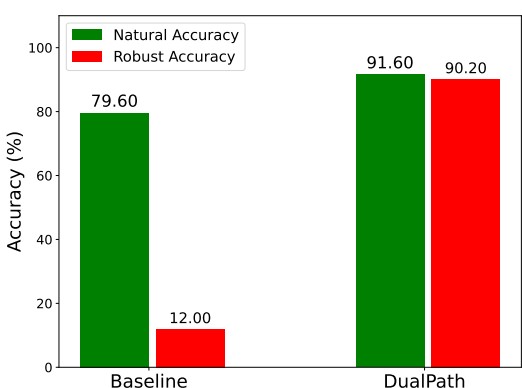

Figure 5: Robustness to blackbox attacks.

Figure 5 shows that DualPath shows remarkable robustness to adversarial examples while the baseline performance drops to almost chance level. This confirms our hypothesis that being less susceptible to spurious correlations in the background and being more object centric considerably enhances the robustness of the model, providing further credence to the utility of incorporating separate pathways for object and scenes in ANNs.

## 6 CONCLUSION AND DISCUSSION

Our study underscores the significant potential of incorporating separate pathways for processing object and scene information within artificial neural networks (ANNs). By emulating the intricate organization of the human brain, this approach demonstrates remarkable performance improvements in model generalization, robustness, and continual learning capabilities. Our empirical analysis highlights how having separate pathways instill several desirable characteristics in the model, including enhanced out-of-distribution generalization and reduced scene bias. By leveraging distinct processing pathways for scene and object information, our approach facilitates the extraction of contextually informed representations, akin to the cognitive mechanisms observed in the human brain. This enables the model to better discern subtle variations in shape, texture, and context, leading to more robust and versatile performance across diverse environments.

However, it is worth noting that we used object masks to create the object and scene images for this study, which may pose practical limitations. One potential solution to this limitation could involve leveraging pretrained foundation segmentation models or foreground extract to automate this process. Additionally, future research could explore more efficient approaches to extracting scene and object information in the representation space and processing them separately, thus further enhancing the scalability and applicability of our proposed framework. Overall, our work highlights the promise of integrating biological insights into AI systems, particularly in the context of scene-object processing. By effectively incorporating separate pathways for scene and object processing, we can develop more cognitively plausible AI systems to address the shortcomings of ANNs.

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

# A    DATASET DETAILS

Here we provide further details of the Tiny-COCO and Tiny-ADE20K datasets used in our study. Figure 6 provides the number of training and test samples for the 10 selected classes. Similarly Figure 7 provides the number of training and test samples for the 12 classes in Tiny-ADE20K. While we attempted to create a more unifrom distribution, these datasets have very high degree of class imbalance and very few instances that could be used for our application. Additionally Figure 8 and Figure 9 provides visual examples of the dataset.

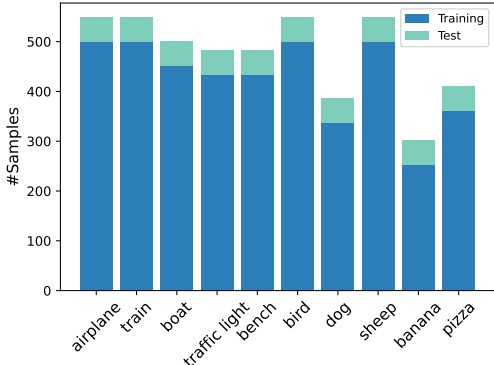

Figure 6: Distribution of training and test samples for Tiny-COCO Dataset.

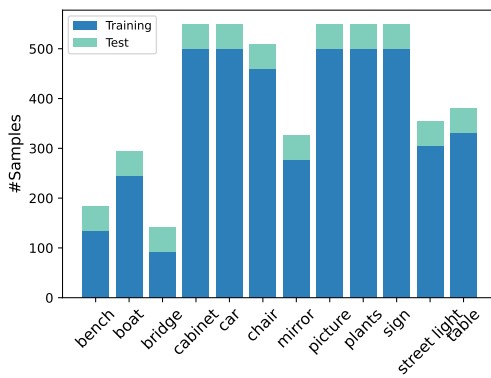

Figure 7: Distribution of training and test samples for Tiny-ADE20K Dataset.

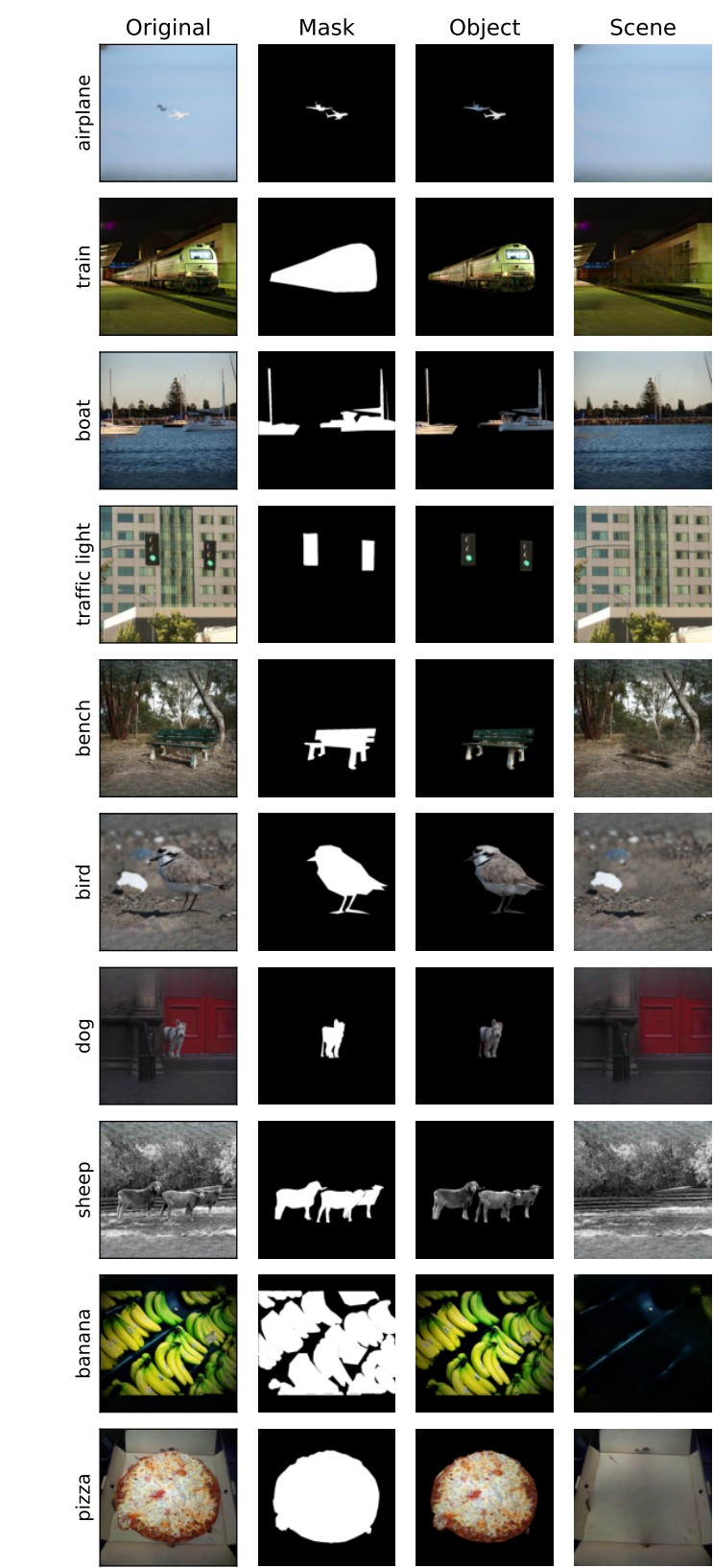

Figure 8: Examples of Tiny-COCO Dataset.

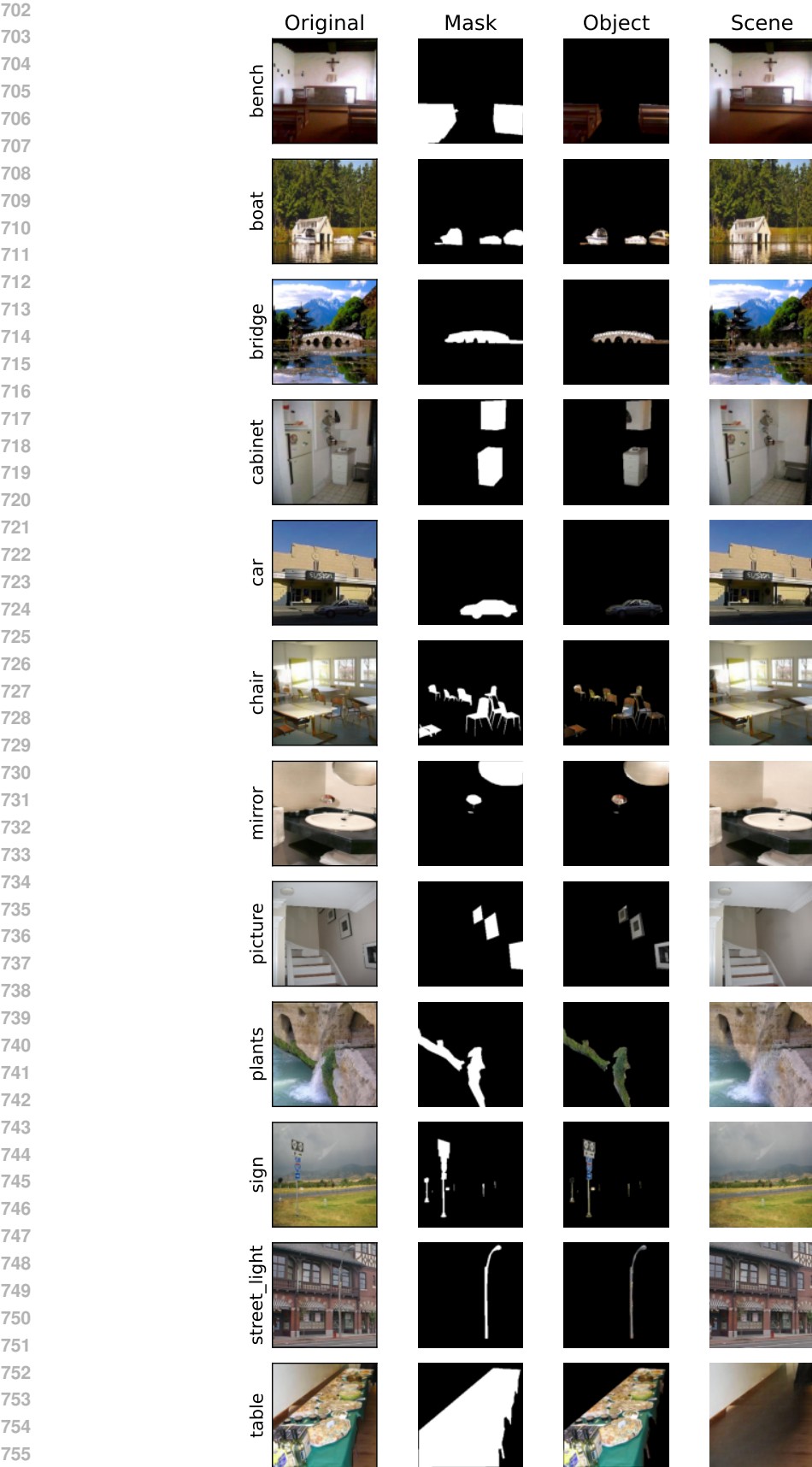

Figure 9: Examples of Tiny-ADE20K Dataset.

