# OpenReview forum: "Dual-Pathway Neural Networks: Harnessing Scene and Object Pathways for Enhanced Visual Understanding"
_ICLR.cc/2025/Conference — ICLR 2025 Conference Withdrawn Submission_

### Official Review · Reviewer_1Rg1 · 2024-11-01

**Soundness:** 1
**Presentation:** 2
**Contribution:** 1
**Rating:** 3
**Confidence:** 3

**Summary:**

The paper introduces DualPath, a dual-pathway neural network designed to emulate the specialized processing pathways in the human brain for scene and object information. DualPath employs dedicated encoders to separately capture scene and object features, followed by a multimodal fusion module that synthesizes these pathways to enhance visual comprehension. This improves object discrimination, cross-domain generalization, and resilience to adversarial challenges over standard ANNs.

**Strengths:**

1. The hypothesis is good.
2. The application to continual learning is promising.

**Weaknesses:**

1. I do not see a clear contribution listed. I still do not understand what is the main focus of the paper. Please list the contributions of the paper clearly. I understand the motivation of the two separate streams, but in terms of performance, please provide quantification of each of the tasks presented in the paper. Please be clear.
2. There are a lot of repetitions in sections 1 and 2, regarding the separation of the object and scene representations and the fused encoder learning. Please mention in one place in a concrete way.
3. I do not see the importance of 2.1 and 2.2. This is already mentioned in Section 1.
4. No related work is given and hence there is a lot of missing literature. Some of the citations are provided.
5. There is no proper block diagram given in terms of methodology. Figure 1 is just a high-level schematic.
6. The adversarial robustness claim is not backed up with experiments.




[1] "A Dual-Stream Neural Network Explains the Functional Segregation of Dorsal and Ventral Visual Pathways in Human Brains.", NeurIPS 2023
[2] "Glance and Focus Networks for Dynamic Visual Recognition", NeurIPS 2020
[3] "Towards Two-Stream Foveation-based Active Vision Learning", IEEE TCDS, 2024

**Questions:**

1. The paper mentions surface level cues at many places. What exactly are these surface cues?
2. On page 1, it is mentioned "ANNs frequently exhibit a propensity to latch onto superficial features... ", please provide some citations.
3. The last sentence of paragraph 1 and the first sentence of paragraph 2 in introduction seem to be contradicting each other. If the ANNs depend more on scene and texture cues, why are they less adept at discerning subtle variations in shape, texture and context?
4. The last sentence in Paragraph 2 can be made a bit more digestible.
5. Why is it multimodal? The information extraction is from the same image. I believe multi-modal is better suited where information is coming from two different modes such as audio and vision, within vision (RGB and Depth). The authors need to come up with a better terminology?
6. In section 2.2, "ANNs lack explicit mechanisms for separate and concurrent processing of scenes and objects...". I think the authors need to clarify that standard CNN classifiers probably do not have this. The authors are not the first one to propose a dorsal and ventral stream based formulation for learning better semantic information. Missing citations. Faster-RCNN, Mask-RCNN, were some initial methods to have shown this effect.
7. To encode the relationships between scene and object, why were multiple-objects per image not considered? This will effect the corresponding baselines as well.
8. For continual learning why only ER has been considered. If 1-2 other baselines should be shown, that might help the claim such as Gradient Projection Memory (GPM).
9. The black-box, threat model is fine. But the system is not tested enough. An epsilon = 8/255 should be the minimum perturbation for a black-box transfer attack setup. Which baseline is used here? What is the surrogate model used to attack baseline and DualPath? The experimental setup is very weak to claim adversarial robustness. This is literally 1 experiment.

---

### Official Review · Reviewer_pBQW · 2024-11-03

**Soundness:** 2
**Presentation:** 3
**Contribution:** 2
**Rating:** 3
**Confidence:** 4

**Summary:**

This work explore the interactions between scene and object and introduce a dual-modality architecture aimed at emulating this cognitive processing mechanism within ANNs.  This approach features separate encodings for scene and object modalities, which are fused to facilitate enhanced visual understanding. By optimizing object recognition and scene reconstruction objectives, our architecture efficiently encodes scene and object information crucial for holistic representation learning.

**Strengths:**

The dual pathways architecture is a research direction that has gained considerable attention in recent years. The design inspiration for this framework primarily comes from the field of neuroscience. The authors provide an understanding of the dual-stream architecture and implement it as a deep neural network (DNN).

**Weaknesses:**

The authors propose this framework with the aim of demonstrating a certain universality. To validate this idea, it is essential to conduct extensive experiments across diverse datasets and applications. Alternatively, they could focus on a specific domain to provide more in-depth insights. Additionally, the implementation approach in this paper diverges from traditional neuroscience experiments, as it utilizes data with segmentation masks, which may affect the generalizability of the findings.

**Questions:**

**Q1:** Bakhtiari et al. (2021) indicate that the functional specialization of the visual cortex emerges from training parallel pathways with self-supervised predictive learning. How can you provide a supplement or further explanation of this work from the perspective of this paper?

**Q2:** As shown in Table 2, the evaluation presented in this paper is quite limited.

Ref:

Bakhtiari, S.; Mineault, P.; Lillicrap, T.; Pack, C.; and Richards, B. 2021. The functional specialization of visual cortex emerges from training parallel pathways with self- supervised predictive learning. Advances in Neural Infor- mation Processing Systems, 34: 25164–25178.

---

### Official Review · Reviewer_KuTv · 2024-11-05

**Soundness:** 2
**Presentation:** 3
**Contribution:** 2
**Rating:** 5
**Confidence:** 4

**Summary:**

This work explores the application of a recent hypothesis from the cognitive science literature that scene-level (background) and object representations are processed independently by dual pathways in visual cortex. The authors use datasets for training that come with segmentation masks such that they can train independent encoders that process segmented objects and the backgrounds separately. In order to turn the backgrounds into realistic images, they use an image inpainter to fill in the image with statistics similar to the rest of the scene. The object pathway is trained for recognition while the scene pathway is trained for reconstruction (to maintain as much context as possible). The encoded representations are then mixed via a simple attention weighting and this representation is trained with cross-entropy loss on the image labels. The final three term objective is used to train both pathways of the network and the authors find that this representation leads to improvements in object recognition (within complex scenes), continual learning, and adversarial robustness.

**Strengths:**

This work addresses an interesting and important question about how global scenes vs. objects are represented in visual processing and whether evidence from neuroscience/cognitive science can help design better deep learning architectures. Specifically I find the idea of implementing a dual pathway architecture for processing scene and objects separately to be interesting and not something I've seen before. Additionally, I like the variety of ways that the new model is evaluated for standard object recognition, lifelong learning, corruptions (blurring), as well as adversarial robustness. In particular I find the adversarial robustness results to be quite interesting as this could be a much cheaper way to achieve robustness than traditional adversarial training. I also find the overall presentation to be relatively clear and well-written.

**Weaknesses:**

1. While I appreciate that the motivation for this work is from Peelen et al. 2024, which provides cognitive/biological evidence for distinct processing pathways for scenes vs. objects, I encourage the authors to not overdo the term "biologically plausible". Very little about the implementation, training, loss functions etc. in this work are biologically plausible or even close to it and there are no evaluations in this work that confirm this learned representation in fact are aligned with biological or cognitive data. As a result, there is very little evidence provided that this network is in fact performing the scene vs. object processing "akin to the cognitive mechanisms observed in the human brain.". I would encourage the authors to state that they are inspired by this coarse idea from the cognitive science literature and hope to verify whether this learned network in fact matches actual cognitive mechanisms- right now I would not make any claims or put emphasis on the plausibility of this specific implementation.

2. This work seems to miss quite a lot of relevant literature about object recognition with contextual scene processing that is quite relevant and potentially should be used as baselines for comparison. Specifically, [1] introduces CATNet which is a two-stream architecture inspired by the ventral / dorsal split which processes one foveal (object-centric) image and a lower resolution global (scene context) in two different pathways. This work was built on in [2] which in fact used transformer encoder/decoder attention layers to do the integration between the scene and object features. While these were trained with a single objective (not the multi-objective used here), the authors in [2] show that their model actually produces similar patterns to human behavior on controlled objects under different contexts. An extension of this work has also shown to model visual search capabilities well (see [3]). Finally, [4] also explores dual stream networks specifically for modeling the ventral and dorsal pathways (one given a small fixation foveal view, and the other a global peripheral view). They train the wide-view network for saliency prediction while the narrow view does object perception and the network representations are fused recurrently to generate the next fixations. While this is a different objective from what is proposed here, I would like this prior work to be discussed.

[1] Zhang, Mengmi, Claire Tseng, and Gabriel Kreiman. "Putting visual object recognition in context." Proceedings of the IEEE/CVF conference on computer vision and pattern recognition. 2020.

[2] Bomatter, Philipp, et al. "When pigs fly: Contextual reasoning in synthetic and natural scenes." Proceedings of the IEEE/CVF International Conference on Computer Vision. 2021.

[3] Ding, Zhiwei, et al. "Efficient zero-shot visual search via target and context-aware transformer." arXiv preprint arXiv:2211.13470 (2022).

3. It seems that there are multiple changes from the baseline that are being made and it it is not clear which components are resulting in the biggest difference. For example, the new two stream architecture now has 2 objectives. The fusion feature and object pathway are trained with the same CE loss as the baseline; however, now the scene encoder is being trained with a reconstruction loss. Is there any evidence that the two streams of DualPath are the critical component or can similar results be achieved in a single architecture but adding a decoder and reconstruction loss? I believe this ablation is critical and would like the authors to do this because it would elucidate whether the main advantage is coming from the additional objective function or the combination of this additional objective **and** the two-stream architecture. Additionally, is the fusion module and scene encoding necessary at all? How well does the object pathway of DualPath perform on it's own? My thought is that if the main gap is due to distractors and background being used in recognition, then shouldn't the object pathway work well even without the scene context fusion? These kinds of ablations are critical to understanding what is important and what is not.

4. Along a similar vain, there is little discussion about the variability of the results relative to the hyperparameters, especially the regularization parameters on the two other loss terms (fuse loss and scene loss). Again this would be helpful to understand as it is unclear right now how specific the results are to the exact tuning of hyper-parameters which can severely limit the practical use-case if these do not generalize to new architectures, training data, etc.

5. For the results in Table 1 I am a bit confused about the evaluation setup. At test time, what images are shown to the object network vs. the scene network. If you are using the ground truth object mask to select the "object image" is this not cheating? Of course, it is hard for the single baseline network to find the object of interest in ADE-20K so it performs poorly, but if you are using the masks to give the DualPath this prior knowledge I don't understand how that is fair. Please correct me if I'm misunderstand what the DualPath network sees at test-time.


If the authors adequately address my concerns I am willing to change my score.

**Questions:**

1. The scene images rely on an inpainting network to fill in the scene. This is a really limiting step to making this practical especially for large datasets.  I understand that this is a proof-of-concept but do the authors have thoughts on how they might be able to implement this in a less expensive way? Perhaps using a transformer and just processing background vs. object tokens?

2. There are plenty of works that are in the realm of object discovery, unsupervised segmentation (slot attention etc.). These methods all work without requiring two streams. Could you achieve similar results simply separating these representations at the output level (in feature space) rather than having to separate the images in pixel space? Or do your results rely on complete separation of the encodings from pixels?

---

### Note · Authors · 2024-11-22

I have read and agree with the venue's withdrawal policy on behalf of myself and my co-authors.